# Structural Assessment of Endodontic Files via Finite Element Analysis

Eyüp Can Kökan [1,*], Abdulkadir Yasin Atik [1], Şebnem Özüpek [1] and Evgeny Podnos [2]

[1] Department of Mechanical Engineering, Boğaziçi University, Bebek, İstanbul 34342, Turkey; ozupek@boun.edu.tr (Ş.Ö.)

[2] Department of Civil, Architectural and Environmental Engineering, University of Texas at Austin, Austin, TX 78745, USA

[*] Correspondence: can.kokan@boun.edu.tr

**Abstract:** A methodology for the structural assessment of Nickel-Titanium (Ni-Ti) endodontic files and a novel approach to predict their fatigue behavior using finite element method (FEM) were proposed. ProTaper-Universal F1 and F2 endodontic files were selected due to availability of extensive test data needed for the validation of the methodology. Bending and torsional loadings were analyzed since these provide essential data for the structural integrity assessment for the endodontic files. High-definition FEM models and their computationally efficient idealized versions were developed. The results for the bending and torsional stiffness of the F1 endodontic file agreed with the literature data validating the proposed methodology. Hysteresis energy density was shown to give promising results as a predictor of low cycle fatigue failure. The predictions with the idealized models matched those of the high-definition models, justifying the proposed idealizations. The validated models demonstrated that F2 has 60% higher bending and torsion resistance and 7% higher hysteresis energy density per cycle with respect to F1, leading to the conclusion that F1 has a lower structural stiffness but a longer fatigue life as compared to F2. In summary, the developed methodology allows for the structural and durability evaluation of various design parameters for Ni-Ti endodontic files.

**Keywords:** root canal treatment; endodontic file; finite element analysis; shape memory alloy





## 1. Introduction

Endodontic instruments that are utilized for root canal treatment might undergo large bending deformations during the procedure due to root canal curvature. Due to their superelastic properties, Nickel-Titanium alloys (NiTi) can withstand large strains without any permanent deformation and are therefore suitable materials for endodontic instruments.

Structural failure of the NiTi endodontic instruments during root canal treatment is not uncommon. Moreover, the understanding of the mechanical response of these files during canal treatment is crucial for selecting the appropriate endodontic file for a specific root canal curvature and operational technique. Consequently, in order to understand the mechanical response and limits of these instruments, several structural assessment test methodologies have been developed, and significant effort has been devoted to measuring and comprehending the structural stiffness and failure behavior of various endodontic file sets. In addition, there has been a significant effort to decrease testing and reduce costs through the development of simulation methods replicating physical tests.

In the context of physical tests, the bending and torsional stiffness of these instruments is evaluated using standardized tests defined by ISO 3630-1 [1]. The ISO standard provides a detailed description of the test apparatus and testing procedure. Several studies have been performed to evaluate the mechanical strength of the ProTaper Universal (PTU) endodontic files, which are the files evaluated in the current study. For instance, Camara et al. compared the flexibility and mechanical response of ProTaper and ProTaper Universal files [2]. Meanwhile, Vieira et al. delved into the effects of clinical use on the torsional behavior of

the ProTaper Universal files [3]. When it comes to measuring the cyclic fatigue durability of endodontic files, although a variety of approaches exist, a specific standard remains elusive [4]. Gambarini et al. developed a cyclic fatigue test apparatus and methodology wherein the endodontic instrument is introduced into a simulated root canal and subjected to rotation until failure [5]. Subsequent studies have adopted Gambarini's methodology to evaluate the cyclic fatigue behavior of diverse endodontic file sets across a spectrum of operational conditions. Fife et al. examined the influence of clinical use of ProTaper endodontic files on their fatigue behavior [6], whereas Whipple et al. compared the fatigue resistance of ProTaper Universal and V-Taper files [7]. In their study, Whipple et al. also investigated the effects of different root canal treatment techniques on the fatigue behavior of the same instrument. Rosa et al. probed the temperature's impact on cyclic fatigue of endodontic instruments [8]. Considering the effects of electromotor torque, Gambarini undertook an investigation employing ProFile instruments [9]. Furthermore, Peng et al. employed a test setup encompassing various curving angles to quantify the fatigue life of ProTaper Universal and ProTaper Next file sets [10].

Examination of the finite element simulations of physical tests reported in the literature demonstrates that numerous studies concentrate on modeling of ISO 3630-1 bending and torsion test scenarios. El-Anwar et al. [11], Santos et al. [12], Prados-Privado et al. [13], and Martins et al. [14] generally outlined the file fixation setup and the load application procedure. They compared and assessed the bending and torsional resistance of the investigated endodontic files by comparing measurements from the test apparatus to results obtained by finite element simulations. Most of these studies provide sufficient information about loading and fixation conditions being simulated but lack comprehensive descriptions of boundary conditions and load application. Furthermore, they omit crucial details concerning the geometry and finite element mesh for both the files and the test apparatus. Additionally, exploration of alternative modeling approaches is notably absent. Within the literature, studies concerning finite element modeling of cyclic fatigue are limited. Lee et al. conducted cyclic fatigue tests on ProTaper Universal endodontic instruments as well as simplified finite element simulations [15]. Ultimately, this study compares failure locations of the files in the tests with peak stress locations derived from the finite element analysis. Similarly, a study by Scattina et al. [16] devised a testing apparatus and conducted durability tests on ProTaper Next files and developed a numerical model which related calculated principal stresses with the measured fatigue life of the file set. In summary, while these studies demonstrate a satisfactory correlation between simulation results and test data regarding the fatigue behavior of endodontic files, they do not report comprehensive modeling details for the test apparatus, file fixation conditions, simulation settings, and contact definitions.

The main goal of this study was to create a comprehensive simulation approach for the bending and torsional tests defined in the ISO 3630-1 standard, as well as for the cyclic fatigue tests introduced by Gambarini [5]. This approach defines required boundary and contact conditions. To enhance computational efficiency and ensure convergence robustness, idealized versions of the simulation models that feature simplified depictions of the testing apparatus were also developed and studied. ProTaper Universal F1 and F2 endodontic files were chosen for this effort, based on the availability of data existing literature.

This study aims to demonstrate the suitability of the developed simulation method for evaluating the structural assessment of endodontic instruments. Additionally, it seeks to establish that hysteresis energy can serve as a reliable parameter for quantifying the fatigue behavior of Nickel-Titanium endodontic files.

## 2. Materials and Methods

In this section, both ISO 3630-1 bending and torsion tests and the Gambarini's cyclic fatigue test configurations are described. Afterwards, finite element simulation of these tests for ProTaper Universal F1 and F2 files are presented. High definition as well as

idealized models are discussed. Finally, the dissipated energy density is evaluated as the fatigue life estimator for Ni-Ti endodontic files.

### 2.1. Description of Physical Tests

#### 2.1.1. ISO 3630-1 Tests

The ISO 3630-1 standard specifies the physical test configurations that aim to characterize the torsional and bending stiffnesses as well as the fracture resistance of endodontic files [1].

#### Torsion Tests

The resistance to fracture by twisting, the angular deflection, and the torsional stiffness of endodontic files are measured with a torque test apparatus.

The apparatus includes a chuck with jaws for clamping the file from its tip, a reversible geared motor for torque application, a torque measuring device, and a chuck connected to the motor where the handle section is mounted (Figure 1).

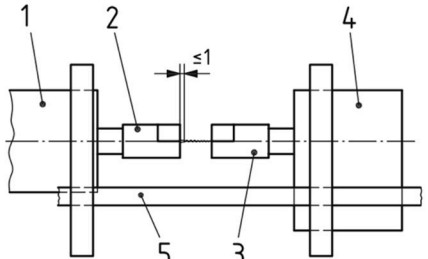

**Key**

1    reversible gear motor
2    chuck with hardened steel jaws
3    chuck with metallic jaws
4    torque measuring device
5    linear ball-bearing

**Figure 1.** Torsion test apparatus [1].

The file handle is placed and tightened into the chuck on the motor side leaving a maximum length of 1 mm to the working section of the file. The tip is placed into the jaws of chuck on the opposite side by 3 mm. The torque is applied from the motor and the file is twisted until it fractures. Meanwhile, the torque measuring device records the angular deflection and the torque applied on the instrument.

#### Bending Tests

The bending test apparatus is utilized to measure the bending stiffness of an instrument. Similar to the torque test apparatus, the bending test apparatus includes a chuck with jaws for holding the file by its tip. Additionally, the system includes a catch pin that is connected to a gear motor and a torque measuring device that measures the torque applied by the motor (Figure 2).

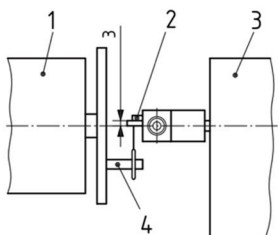

**Key**

1    reversible gear motor
2    stop
3    torque measuring device
4    catch pin

**Figure 2.** Bending test apparatus [1].

The file tip is placed into the jaws of the chuck by 3 mm and tightened. The handle is kept free. The motor is rotated such that the catch pin comes into contact with the file handle. After contact, torque is applied by the motor and the catch pin is rotated 45 degrees, subjecting the tested endodontic file to bending. The torque measurement device records the rotation and the applied torque.

### 2.1.2. Cyclic Fatigue Tests

The test apparatus utilized for the cyclic fatigue life assessment of the endodontic files (Figure 3) contains artificial canals with various curvatures. The endodontic file is inserted into these canals and rotated until failure. The cyclic fatigue behavior of the file is characterized in terms of the failure location and the number of rotations to fracture.

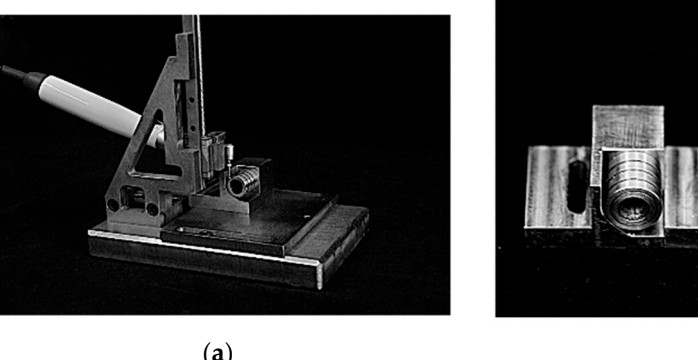

(**a**)　　　　　　　　　　　　　　　　　(**b**)

**Figure 3.** Test apparatus proposed by Gambarini [10]. (**a**) Apparatus with the file inserted into the artificial canals. (**b**) Photo of the apparatus in cross-section.

The apparatus is composed of two main components, a steel cylinder with grooves and canals and a steel jig. The 1 mm depth and 1.5 mm width canals are formed between the grooves of the cylinder. The radii of the cylinder and the jig determine the canal geometry, whereas the angle ($\alpha$) of the jig determines the canal curvature (Figure 4).

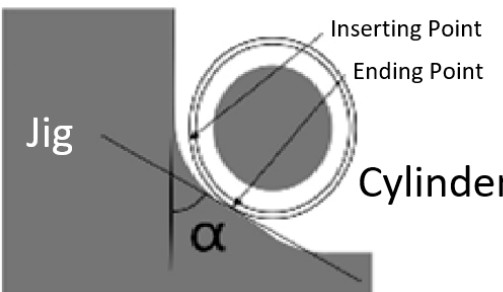

**Figure 4.** Test apparatus cross-section schematics [10].

In literature, it is common for the simulated canal to have a 5 mm radius and 90 degrees curvature; these measurements were used in this work. To create this canal geometry, a cylinder with 6 mm radius and 1 mm deep grooves and a jig with 6 mm radius are utilized. After such a configuration is arranged, the cylinder and the jig are positioned so that the file insertion location has a 2 mm width and the file end point has a 1 mm width. The jig surface is extended 13 mm straight vertically and horizontally before the curvature begins and after it ends (Figure 5).

The handle of the file is attached to an electric motor and the file is inserted 22 mm from the top of the jig. The file is positioned so that its middle section is in contact with the steel cylinder surface, whereas the tip of the file is located around the file end point and is in contact with the jig.

After positioning, the endodontic file is rotated with the aid of the electric motor until failure. During the process, the file is constantly air cooled to maintain room temperature and lubricated by oil to reduce friction [10].

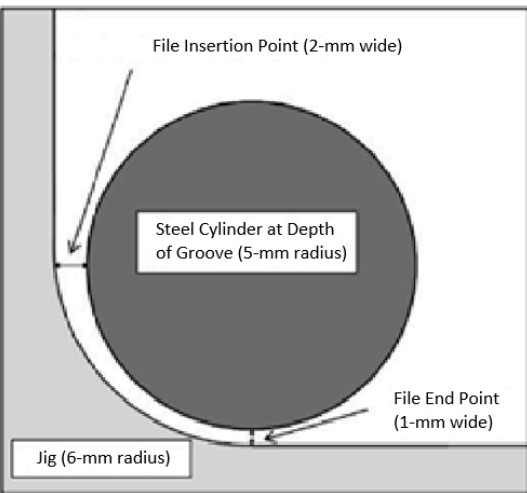

**Figure 5.** Simulated canal geometric parameters [7].

## 2.2. Finite Element Modeling of Physical Tests

Physical tests described in the previous section are modeled via FEM. In the finite element model development, high-definition representation of the test configuration is emphasized. The computed deformations, stress, and dissipated energy are compared to the available data reported in the literature, as presented in the following sections. In addition, idealized versions of the models are also developed to reduce computational time and improve convergence behavior. Simulations are performed in Abaqus [17]. Since exactly the same methodology is used for simulations involving F1 and F2 endodontic files, the details of FEA model development are only reported for F2 endodontic file.

Material properties of conventional endodontic files (those without special heat treatment) are considered for the simulations. In particular, properties measured by Santos et al. [12] from tensile tests of 1 mm Nitinol wires are used (Table 1).

**Table 1.** NiTi properties.

| | |
|---|---|
| Austenite Elasticity | 42,530 MPa |
| Austenite Poisson's Ratio | 0.33 |
| Martensite Elasticity | 12,828 MPa |
| Martensite Poisson's Ratio | 0.33 |
| Transformation Strain | 10% |
| $(\delta\sigma/\delta T)$ Loading | 6.7 |
| Start of Transformation Loading | 492 MPa |
| End of Transformation Loading | 630 MPa |
| Reference Temperature | 22 °C |
| $(\delta\sigma/\delta T)$ Unloading | 6.7 |
| Start of Transformation Unloading | 192 MPa |
| End of Transformation Unloading | 97 MPa |
| End of Martensitic Elastic Regime | 1200 MPa |

The built-in shape memory alloy (SMA) material implementation in Abaqus is based on Auricchio's formulation which defines the austenite, martensite, and transformation phases [17]. This implementation also accounts for the effect of temperature [18,19], represents the inner hysteretic loop effect and accurately captures the maximum change in tensile strain, which is essential for accurate prediction of the fatigue life of a shape memory alloy [20].

### 2.2.1. Meshing and Solver Convergence Criteria

During FEM mesh development, the element quality criteria reported in Table 2 were enforced to maintain the accuracy of the results, convergence behavior, and numerical stability.

**Table 2.** Element quality criteria.

| Quality of Brick and Penta Elements | | |
|---|---|---|
| Warpage | 95% < 40° | 5% < 50° |
| Aspect Ratio | 95% < 10 | 5% < 20 |
| Skew | 95% < 70 | 5% < 80 |
| Minimum angle | >45°<br>(min. 90% > 45° and 10% > 20°) | |
| Maximum angle | <145°<br>(min. 90% < 135° and 10% < 165°) | |
| Jacobian | 95% > 0.4 | 5% > 0.25 |
| Quality of tetra elements | | |
| Tetra Collapse | 0.13 (minimum 95% > 0.3 and 5% > 0.2) | |

The default convergence criteria of Abaqus, are applied for the FEA models [20].

### 2.2.2. Finite Element Mesh Development for ProTaper Universal F2 Endodontic File

The process of acquiring the endodontic file's geometry began with a 3D scan of the file within a micro computed tomography (MicroCT) scanner Skyscan 1275, accompanied by the utilization of 3D reconstruction software NRecon (V1.7.4.6) [21]. The scanner generated images of the endodontic file's geometry in TIFF format, and subsequently, the NRecon software was employed to transform these TIFF images into BMP format. The next stage involved the utilization of open-source software 3D Slicer (V4.11.20210226) [22] to combine the BMP images into a single STL geometry data file that defined the resultant 3D surface (Figure 6).

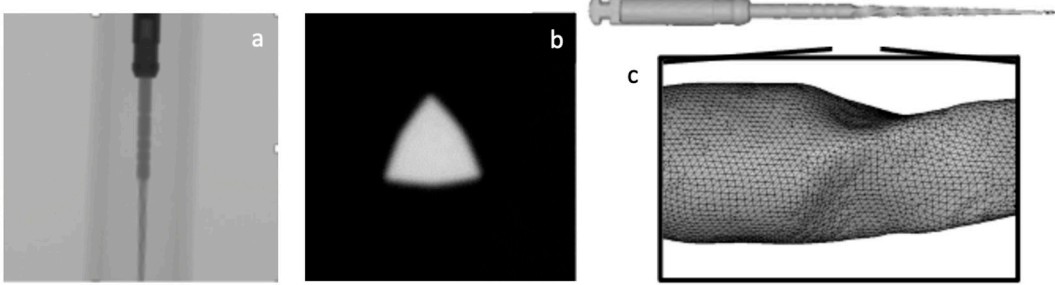

**Figure 6.** F2 File Geometry Scanning Steps: (**a**) An illustration of geometry stored as a TIFF image; (**b**) An illustration of geometry stored as a BMP image; (**c**) STL-triangulated surface geometry stored as a single STL geometry.

The final STL file contained the 3D surface of the endodontic file as defined by triangular elements. Once the STL geometry file was generated, it was imported into the finite element pre-processor software and subjected to the following examination:

Identification of free surface edges arising from potential imperfections in the scanning process.

Assessment of geometric details that could be omitted from the analysis to reduce the model's size and reduce the analysis time.

After the geometry was reviewed and corrected, AutoDesk Fusion 360 [23] was employed to derive surface data from the STL geometry file. In the final phase, the resultant surface geometry of the endodontic file was partitioned into 40 segments by parallel planes perpendicular to the endodontic file's axis, and the cross sections were defined. Finally, a 2D surface mesh was constructed on each cross section and swept along the endodontic file's axis to form solid elements (Figure 7).

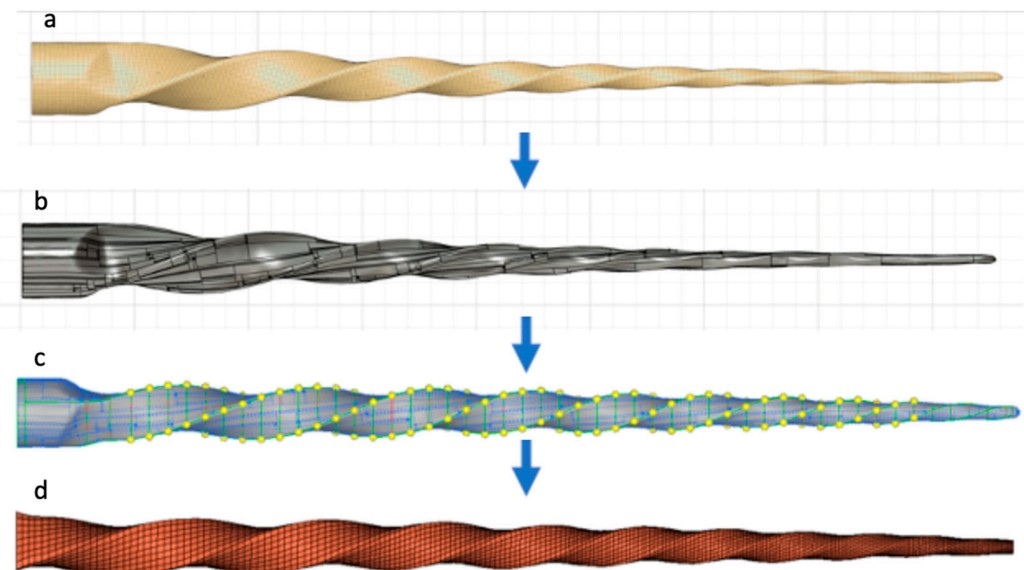

**Figure 7.** F2 file geometry processing stages: (**a**) STL representation; (**b**) Surface geometry defined by AutoDesk Fusion 360; (**c**) 3D surface partitioned by parallel planes; (**d**) Final 3D solid mesh.

### 2.2.3. ISO 3630-1 Test Modeling

Bending Test Modeling

In this section, a high-definition bending test simulation (Figure 8) incorporating the file as well as the catch pin and the idealized versions (Figure 9) of the model where the catch-pin is simulated by force boundary conditions are presented and evaluated. The catch pin is meshed with first order 3D brick elements. Hard contact is defined between the pin and the file handle without tangential friction.

- Boundary conditions for the high-definition model:

All degrees of freedom for the surface nodes within 3 mm from the tip of the endodontic file are constrained kinematic coupling elements, and the pin is connected to the center of rotation via kinematic coupling elements [17]. Following the initial contact with the file handle, the catch-pin is rotated 45 degrees around the rotation axis.

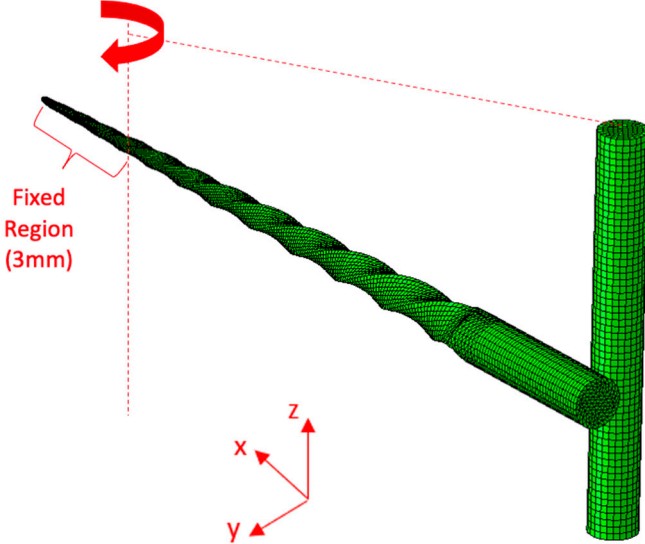

**Figure 8.** Bending test high-definition model.

- Boundary conditions for the idealized model

In order to simulate the catch-pin replacement, a force is applied at the independent node of a kinematic coupling element, which is located at the handle axis where the initial contact with the catch pin is observed. The force follows the nodal rotation. The simulation ended when the rotation of the handle reached 45 degrees.

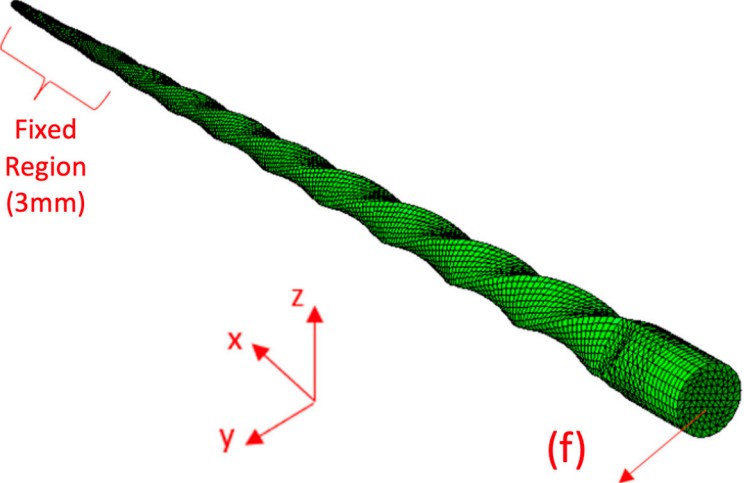

**Figure 9.** Bending test idealized model.

Torsion Test Modeling

The torsion test model consists only of the endodontic file (Figure 10).

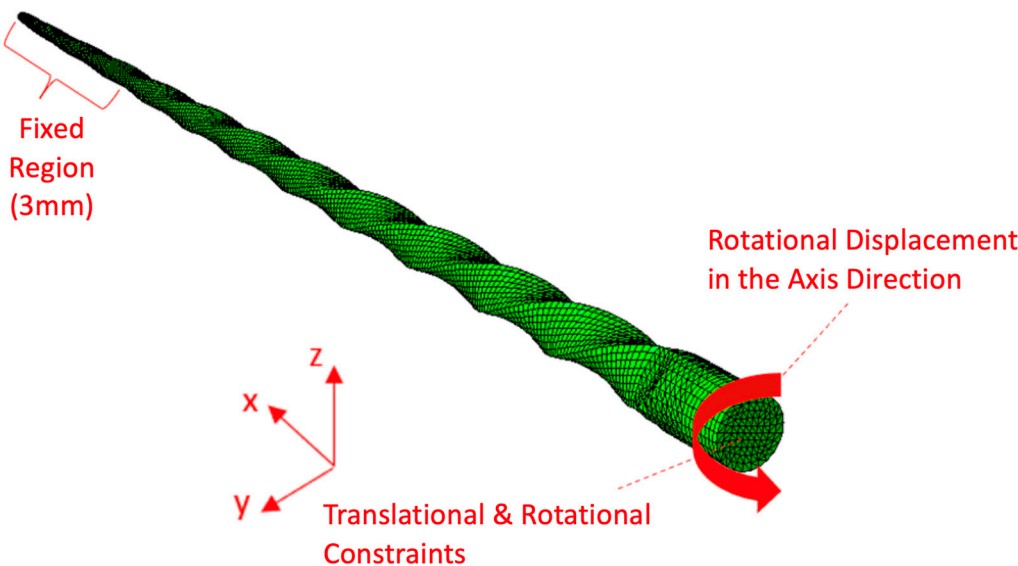

**Figure 10.** Torsion test high-definition model.

- Boundary conditions for the high-definition model

Surface nodes of the endodontic file with 3 mm of the tip are fully constrained. Constraints are imposed on the handle section using a kinematic coupling element, restricting its translational and rotational degrees of freedom other than the axial rotational direction, where rotational displacement is defined. Twisting motion on the handle is simulated through rotational displacement applied to the kinematic coupling element. This rotational displacement is applied until the twisting moment reaches 3 Nmm, following the approach by de Arruda SL et al. [12].

- Boundary Conditions for Idealized model

The displacement constraints at the handle side are removed, allowing the handle unrestricted movement in all directions.

### 2.2.4. Cyclic Fatigue Test Modeling:

- The high-definition model

The model includes an endodontic file where the jig part has 90 degrees curvature and 6 mm radius and the cylinder part has a groove of 5 mm radius and 1.5 mm width. The linear elastic material model for steel with 210 GPa Young's modulus and 0.3 Poisson's ratio is used for the jig and cylinder. The cylinder and the jig are modeled with first order brick elements, whereas the file is modeled as described in Section 2.2.2 (Figure 11).

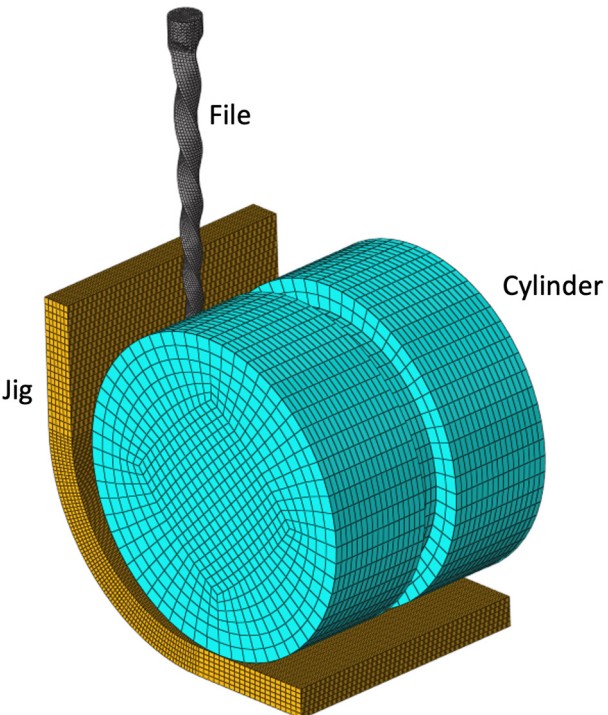

**Figure 11.** Cyclic fatigue test configuration model.

The prescribed displacement boundary conditions are applied to the endodontic file's end surface via a kinematic coupling element. The jig and the cylinder's outer surfaces are fixed.

- The Idealized model

Considering that the test apparatus components (jig and cylinder) are made of steel, which is 5 times stiffer than NiTi, the deformation of the apparatus is estimated to be negligible as compared to that of the endodontic file. Consequently, the simulated canal formed by the apparatus geometry is simulated by analytical rigid surfaces (Figure 12).

The finite element simulation involved the following steps:

1. Endodontic file insertion: The file is inserted to a depth of 22 mm from the top of the jig.
2. Endodontic file rotation: After insertion, the file's axial position is fixed, and it is rotated around its axis until the dissipated energy stabilizes.

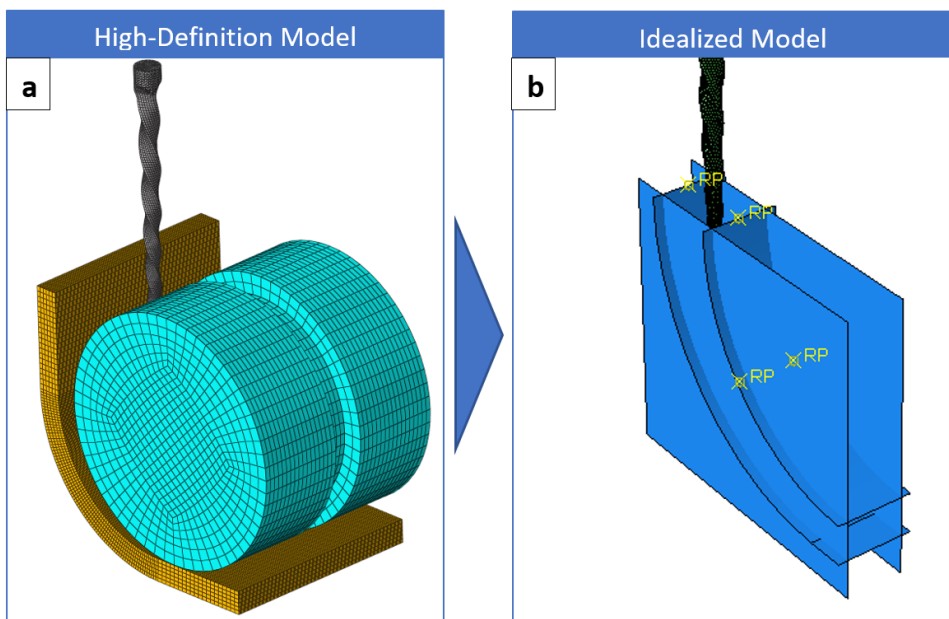

**Figure 12.** Cyclic fatigue test configuration model. (**a**) High-Definition Model, (**b**) Idealized model.

#### 2.2.5. Dissipated Energy Density for Fatigue Life Estimation

Moumni et al. asserted that the hysteresis energy per cycle holds significance in estimating the low cycle fatigue life of shape memory alloys due to the stabilizing trend observed in the hysteresis curve after multiple cycles [24]. Specifically, in the case of uniaxial loading, the count of cycles until failure is determined by the dissipated energy as measured using hysteresis curves. Moumni et al. have validated this proposed approach in scenarios involving torsional loading [25]. Furthermore, Gu et al. have extended this methodology to cases of multiaxial loading [26].

Based on the findings from the reviewed literature, the dissipated energy proves to be a convenient parameter for assessing the fatigue life of endodontic files, which are commonly subjected to intricate loading patterns. It is important to note that different endodontic file sets are typically manufactured from various NiTi alloys and undergo different heat treatments. Hence, establishing a calibrated relationship between dissipated energy and fatigue life is necessary. The computed hysteresis energy can also be used to compare different endodontic file designs made from the same material or evaluating various surgical techniques applied to an identical configuration.

In the current study, the dissipated energy density, also referred to as 'hysteresis energy density,' has been employed to predict the overall fatigue behavior of the F1 and F2 endodontic files and identify the location of failure. Currently, the hysteresis energy is used for comparison purposes, while the calibration of the relationship between dissipated energy and fatigue life is not within the scope of this study.

### 3. Results

This section presents the results for the described models for the ISO 3630-1 bending and torsion tests, as well as the cyclic fatigue test. The results of the high definition and idealized models are compared with each other. Additionally, for the ISO 3630-1 tests, F1 endodontic file results are also correlated with the test data reported in the literature. For the cyclic fatigue simulations using idealized canal configuration, the results are correlated with the test data reported in the literature for both the F1 and F2 endodontic files.

#### 3.1. ISO 3630-1 Test

3.1.1. Bending Test

Regarding the F1 file, in the high-definition model with a catch-pin, the handle tip displaces 0.956 mm (Figure 13), underestimated as 0.002 mm (Figure 14) in the idealized

model. Maximum von Mises stresses are similar, differing by at most 0.9% (Table 3). Bending moments are identical in both high definition and idealized models with excellent agreement with the test data reported in Arruda SL et al. [12] (Figure 15).

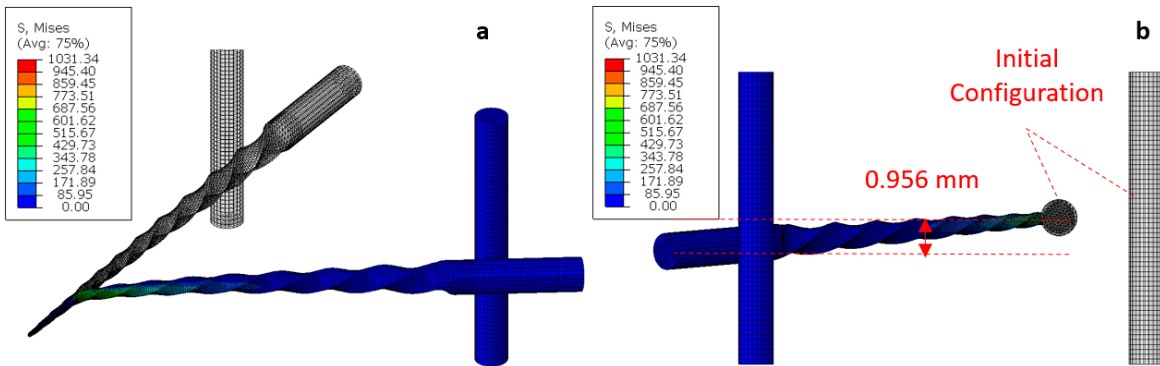

**Figure 13.** F1 File bending test high-definition model results. (**a**) Isometric View, (**b**) View in the axial direction.

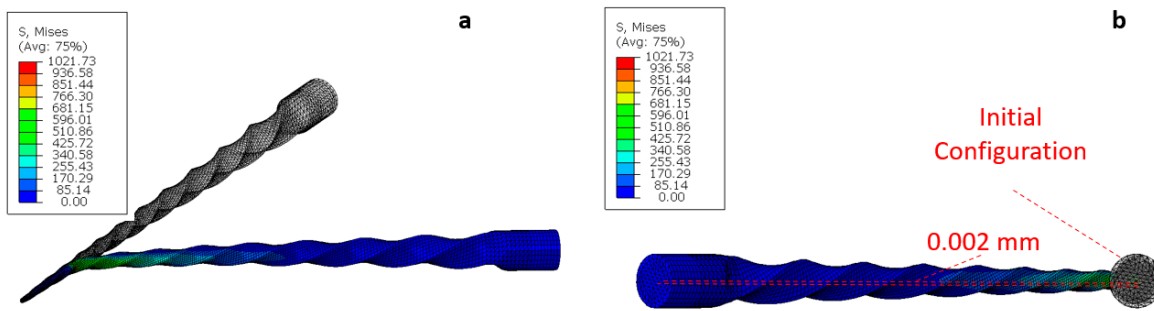

**Figure 14.** F1 File bending test idealized model results. (**a**) Isometric View, (**b**) View in the axial direction.

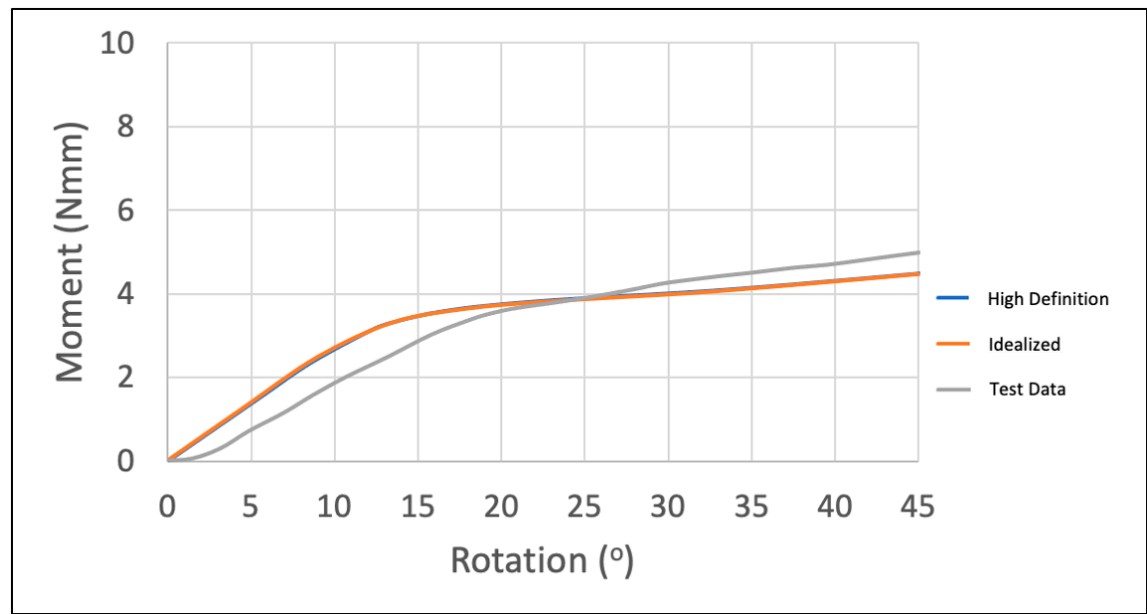

**Figure 15.** Bending moment for F1 Endodontic File.

A similar trend is observed for the F2 endodontic file. In the high-definition model, the handle tip displaces by 0.473 mm, while the idealized model underestimates it at 0.229 mm. Maximum von Mises stresses differ by around 2.4% (Table 3), and bending moments match in both models with 7.85 Nmm at 45 degrees of bending angle.

**Table 3.** Von Mises Stresses for F1 and F2 endodontic files bending simulation.

| Maximum von Mises Stresses | High-Definition Model | Idealized Model | Difference |
|---|---|---|---|
| PTU F1 | 1031.34 MPa | 1021.73 MPa | 0.9% |
| PTU F2 | 1211.20 MPa | 1239.91 MPa | 2.4% |

### 3.1.2. Torsion Test

Results indicate that, for the F1 endodontic file, maximum von Mises stresses are 27% higher than those for the F2 file (Figure 16). Regarding the handle rotation, it is observed that, by the end of the simulation, the F1 handle rotates 68% more than the F2 file handle. It is also observed that the handle rotation prediction of F1 endodontic files is in excellent agreement with the test data provided by Arruda SL et al. [12] (Figure 17).

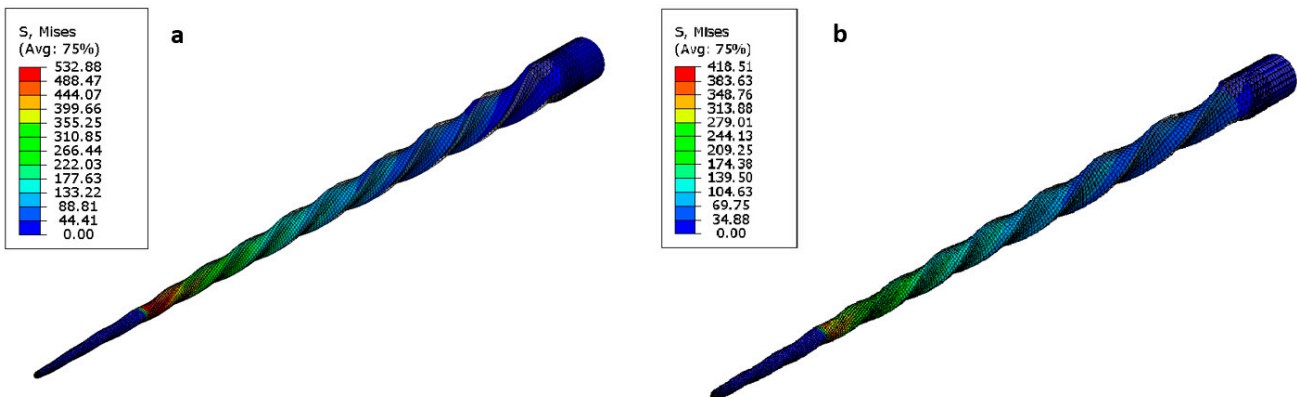

**Figure 16.** F1 and F2 File torsion simulation results for F1 and F2 endodontic files. (**a**) PTU F1, (**b**) PTU F2.

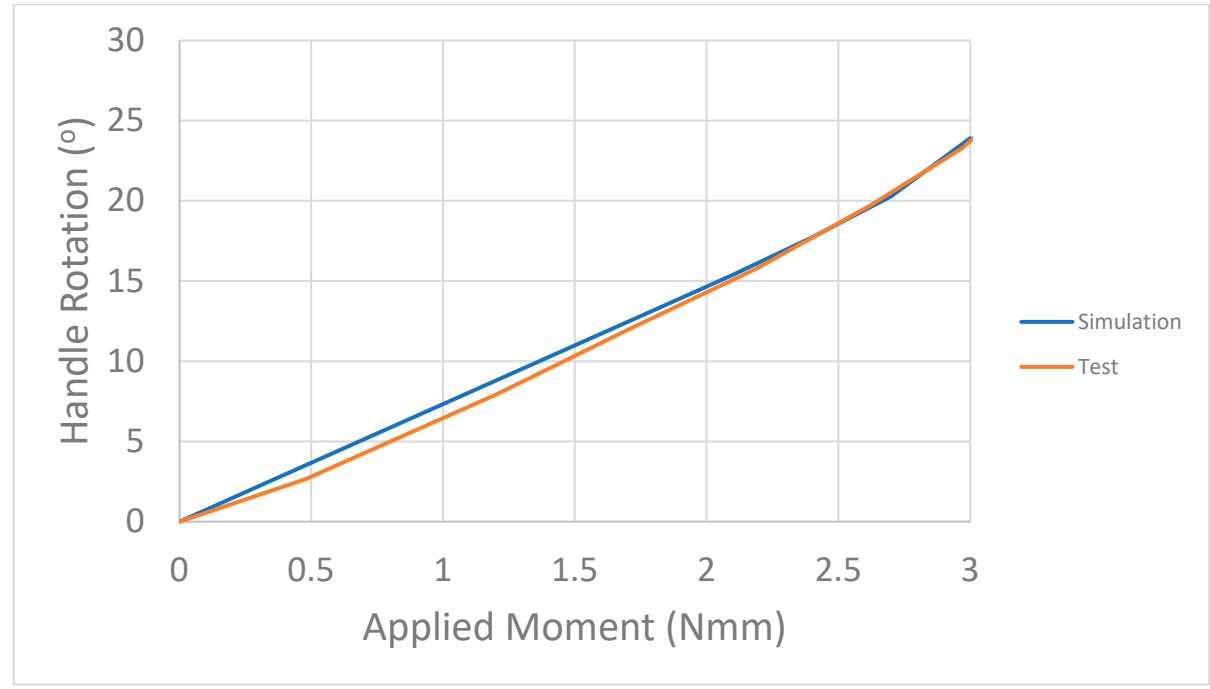

**Figure 17.** F1 endodontic file torsion simulation results.

### 3.2. Cyclic Fatigue Test

The cyclic fatigue simulation, including the canal, shows similar deformation behavior for both endodontic files. At the end of the insertion step, the endodontic file's mid-section contacts the cylinder at Point A, while the file's tip contacts the jig at Point B, which is located where the jig's cylindrical and horizontal flat surfaces meet (Figure 18).

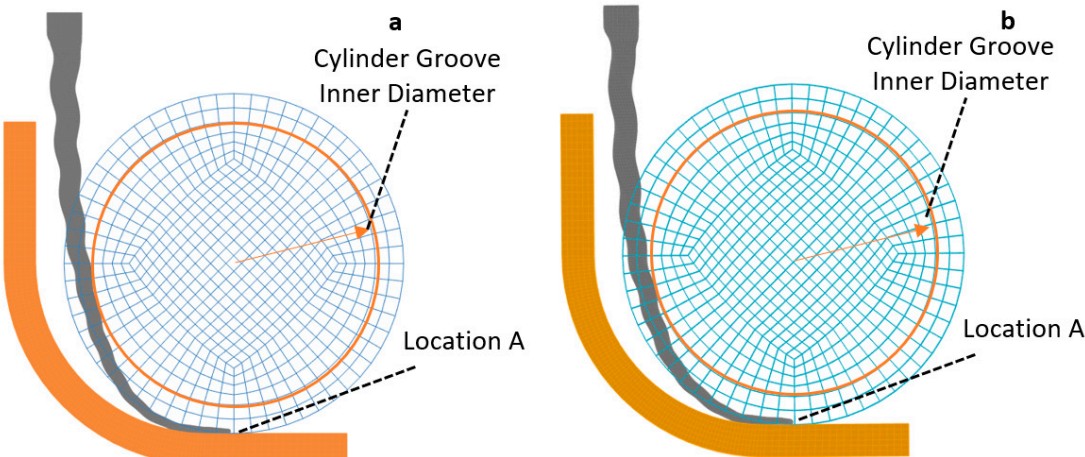

**Figure 18.** The deformed configuration of endodontic files in Gambarini's test apparatus at the end of Insertion Step (**a**) PTU F1, (**b**) PTU F2.

The highest hysteresis energy density is observed at 4.1 mm from the tip of the PTU F1 endodontic file (Figure 19), which corresponds to the expected failure location. For the PTU F2 file, the simulation results indicate a failure location of 3.48 mm from the tip (Figure 20).

In terms of fatigue life estimation, a maximum dissipated energy density of 64.0 MJ/m$^3$ was computed for the F1 endodontic file (Figure 19), while for the F2, it was 68.5 MJ/m$^3$ (Figure 20).

Overall, the idealized model, in which the simulated canal is created using analytical surfaces, yields identical results to the high-definition model, confirming its validity.

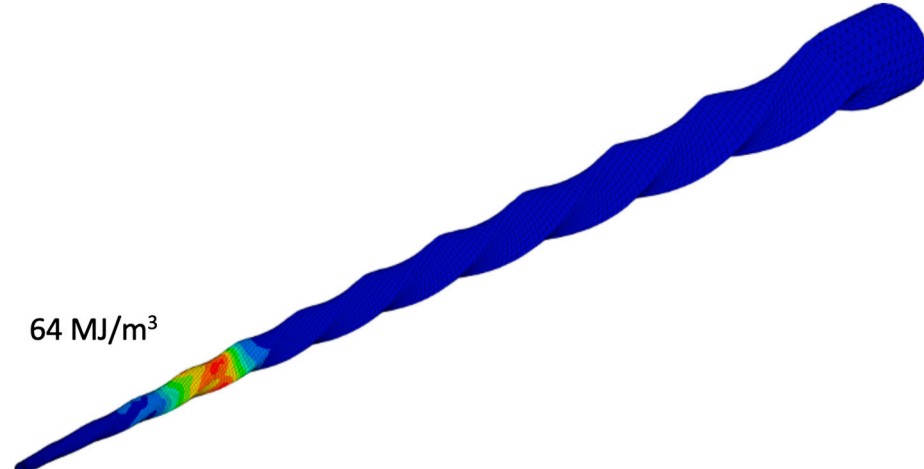

64 MJ/m$^3$

**Figure 19.** Maximum dissipated energy density in F1 file.

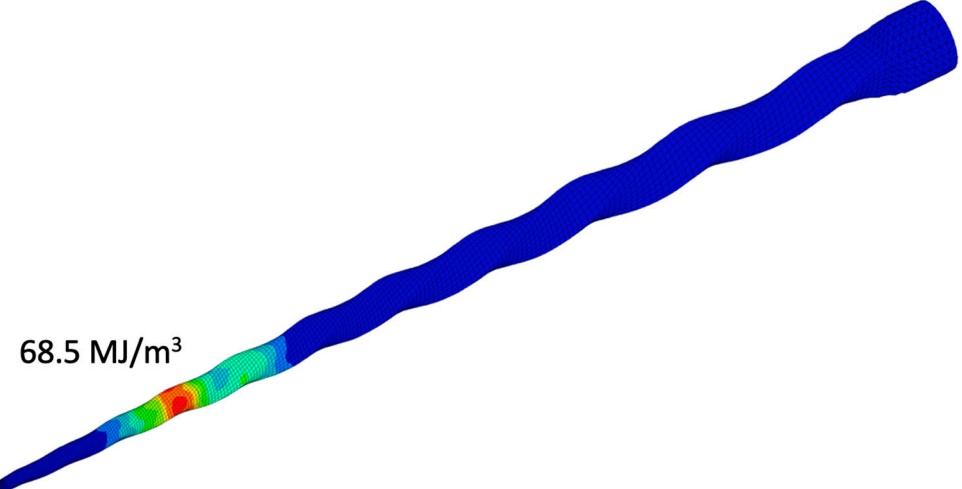

**Figure 20.** Maximum dissipated energy density in F2 file.

## 4. Discussion

- ISO 3630-1 Simulations

The high-definition model described for the bending test demonstrates results that exhibit a reasonable correlation with the physical test outcomes reported by de Arruda SL et al. [12], particularly concerning the bending moment estimates for the F1 endodontic file. This agreement extends to the torsion tests, where the simulation outcomes closely align with the test results, reported by de Arruda SL et al. [12]. Furthermore, the simulation results confirm that the outcomes obtained from the idealized models show a high level of accuracy when compared to the results from the high-definition models. While the horizontal displacement of the handle due to the geometric stiffness of the file is underestimated by the idealized models, it does not impact the calculated bending moment results. In summary, this validation confirms that the proposed methodology using high-definition models is suitable for assessing the structural stiffness of the endodontic files, and the suggested idealizations are indeed appropriate.

Another noteworthy result is that the F1 endodontic file has approximately 57% less torsional stiffness and 68% less bending stiffness compared to the F2 file. Again, these findings align with the test data presented by Camara et al. [2].

- Cyclic Fatigue Simulations

The simulation results for the Gambarini's test system indicate that, during the insertion step, the mid-section of the file touches the cylinder while the tip of the file contacts the jig's cylindrical and horizontal surfaces at their junction. This pattern is consistent with the observations made by Fife et al. in [6]. In the same study, focusing on the F2 endodontic file, Fife has reported an average failure location of 3.5 mm from the file's tip. The present simulation predicts the highest dissipated energy density location of 3.48 mm from the tip, which corresponds to this study. For the F1 endodontic file, Fife reported a failure location between 4.0–5.8 mm, a range that matches the simulation's prediction of 4.1 mm from the tip.

For fatigue life estimation, Fife has recorded an average of 367 rotations for the F1 endodontic file and 320 rotations for the F2 endodontic file before failure. Simulation results indicate 8% higher dissipated energy for the F1 file compared to the F2 file in the most critical area, pointing to a longer fatigue life for endodontic file F1. The validation further demonstrates that hysteresis energy serves as a reliable parameter for quantifying the fatigue behavior of Ni-Ti endodontic files.

Similar to ISO 3630-1 test models, in principle, the idealized models are observed to have better numerical convergence behavior with increased computational efficiency.

It should be noted that the proposed methodology has certain limitations due to the utilization of the Abaqus SMA software (Abaqus 2021) module in this study [17]. While the module is well-suited for the methodology proposed in this work, particularly due to its ability to represent the inner loop effect, and its suitability and accuracy which are validated and verified with the test data, it employs a symmetric material properties implementation that treats tension and compression equally. However, an asymmetric implementation would likely result in more accurate overall results. Such an implementation could be achieved through the use of a UMAT (User Material) subroutine and would necessitate detailed material property data.

## 5. Conclusions

In this paper, the physical tests utilized for the structural assessment of the endodontic instruments are reviewed. Finite element-based simulation methodology for these test configurations is proposed. Simplification of the models and various phenomena related to deformational behavior of the files are also discussed. The proposed methodology is validated against test data reported in literature.

ISO test simulation results indicate that the PTU F1 endodontic file has lower torsional and bending resistance. It's also observed that the PTU F1 endodontic file is expected to have longer fatigue life compared to PTU F2 endodontic file; it also tends to have a failure location further from the tip compared to PTU F2 file. These results are supported by the test data reported in the literature.

In this work, the dissipated energy is assumed to be a relevant parameter for fatigue life estimation and is utilized to evaluate the fatigue behavior of the ProTaper Universal endodontic files. Estimating the complete fatigue lifespan of an endodontic file could become feasible by calibrating the mathematical model proposed by Moumni [23] to the results of cyclic tensile-compression tests performed on the specific NiTi alloy employed in each file. This calibration is regarded as a potential avenue for future research.

**Author Contributions:** E.C.K. conceptualized and conducted the study (as part of his PhD thesis), and wrote the original manuscript. A.Y.A. developed the image processing and geometry acquisition methodology. Ş.Ö. and E.P. provided research guidance and contributed to the definition of models and to reviewing the analysis results. All authors have read and agreed to the published version of the manuscript.

**Funding:** This research was funded by Boğaziçi University, Grant Number 18582D.

**Institutional Review Board Statement:** Not applicable.

**Informed Consent Statement:** Not applicable.

**Data Availability Statement:** Not applicable.

**Acknowledgments:** The authors gratefully acknowledge the financial support of the Boğaziçi University Research Fund (Grant Number 18582D). The authors would also like to extend their gratitude to Raif Erişen and T. Yağmur Yalçın for many useful discussions throughout the course of this work.

**Conflicts of Interest:** The authors declare no conflict of interest.

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
