# Peer review of "Structural Assessment of Endodontic Files via Finite Element Analysis"

_applsci, doi:10.3390/app131810293_

Round 1
Reviewer 1 Report
ABSTRACT
A brief introduction explaining why the examination of ProTaper-Universal F1 and F2 files and their qualities is essential in the field of endodontics would be beneficial.
The abstract indicates that "test system properties" are identified and a "methodology" for complete representation is established, but no specifics regarding these techniques are provided.
The abstract says that the comprehensive models produce findings comparable to experimental data from the literature, however it does not specify where this data came from.
The abstract focuses primarily on Shape Memory Alloy (Nickel-Titanium) ProTaper-Universal F1 and F2 files. While this restricted scope may be appropriate for a given study topic, it restricts the findings' broader relevance and generalizability.
The abstract says that simulations show that the complete models produce findings that are comparable to experimental data, however it does not give particular numerical numbers or statistical comparisons. Incorporating some crucial quantitative data might improve the abstract's readability.
The abstract lacks clear conclusions or implications of the results.
INTRODUCTION
While the study's principal goal is addressed, it may be conveyed more simply and concisely. To improve readability, the phrases might be split down into smaller, more focused assertions.
The formatting of the literature citations is inconsistent, which might be irritating for readers.
The introduction includes a number of prior research but does not provide a clear organisation or synthesis of the current knowledge.
While the introduction acknowledges earlier research, it does not critically examine or emphasise their flaws.
Some information in the introduction is repeated, resulting in duplication. By reducing redundant duplication of concepts, the writing might be simplified.
The study's research topic or hypothesis is not stated directly in the introduction.
While the introduction discusses the mechanical reaction and limitations of endodontic tools, it does not go into detail about why understanding these features is important in the discipline of endodontics.
Some terminology, such as "structural assessment test methodologies" and "model simplification investigations," are poorly defined, leaving readers in the dark regarding their precise meaning.
METHODOLOGY
The Methodology part is fairly lengthy and includes several components of the study, such as physical test descriptions, finite element mesh construction, and fatigue life estimation. Subdividing the section would make it more organised and simpler for readers to follow.
There is some matter that is repeated between subsections. The presentation of the complete bending test model, for example, is repeated in the section on finite element modelling. To make the writing more streamlined, reduce duplication.
Although the approach states that "comprehensive representation of the test configuration is emphasised," detailed data concerning the boundary conditions, mesh settings, and convergence criteria are omitted. These particulars are critical for readers to comprehend and duplicate the simulations.
The method covers the use of dissipated energy density for estimating fatigue life, however it does not go into detail about how this relationship was calibrated.
RESULTS
Too lengthy and confusing the authors should simplify this section.
There are too many figures making it very confusing the authors should make a collage and submit as (a, b, c, etc).
Reviewer 2 Report
Dear Authors,
Please find below some observations and recommendations concerning your article entitled” Structural Assessment of Endodontic Files via Finite Element Analysis”. I appreciate your efforts and insights into the subject matter.
Title and authors
Please decide the corresponding author (*).
In the Abstract section:
The abstract of your article needs some restructuring. It would be beneficial to rewrite it without the use of bullet points, while ensuring that it succinctly highlights the key points of your study.
In the Introduction section:
The introduction should be more focused on introducing the subject matter to the readers. It's essential to provide context and rationale for your study in a clear and concise manner.
In the Materials and Methods section:
This section requires attention to detail, particularly in subchapter numbering. Clarity and coherence are crucial here, so a careful rewriting is recommended. Additionally, when presenting figures and tables, be mindful of any potential overlapping issues that may affect readability.
In the Results section:
I kindly suggest that you present your study results without directly comparing them to the literature. This section should focus solely on the outcomes of your study.
In the Discussion section:
The discussion section is an appropriate place to make comparisons between your results and existing literature. It's also an opportunity to provide explanations for the data you've obtained.
Please ensure that there is no duplication of information between different sections of the article ("Considering the fatigue life estimation, Fife et.al [3]...." ).
Also, a paragraph highlighting the limitations of your study should be added here.
In the References section:
Please follow the styles recommended for MDPI journals.
Round 2
Reviewer 1 Report
Appreciate the authors for revising the manuscript. all my concerns are addressed. Now the paper is much better in the flow and content.
Reviewer 2 Report
Dear authors,
I want to extend my appreciation for the revisions made to the manuscript. You have successfully addressed all of my concerns, resulting in a significant improvement in both the paper's coherence and content.